# Increasing Uptake of COVID-19 Vaccination and Reducing Health Inequalities in Patients on Renal Replacement Therapy—Experience from a Single Tertiary Centre

**DOI:** 10.3390/vaccines10060939

**Published:** 2022-06-13

**Authors:** Dimitrios Poulikakos, Rajkumar Chinnadurai, Saira Anwar, Amnah Ahmed, Chukwuma Chukwu, Jayne Moore, Emma Hayes, Julie Gorton, David Lewis, Rosie Donne, Elizabeth Lamerton, Rachel Middleton, Edmond O’Riordan

**Affiliations:** 1Department of Renal Medicine, Northern Care Alliance NHS Foundation Trust, Salford M6 8HD, UK; rajkumar.chinnadurai@nca.nhs.uk (R.C.); chukwuma.chukwu@nca.nhs.uk (C.C.); jayne.moore@nca.nhs.uk (J.M.); emma.hayes@nca.nhs.uk (E.H.); julie.gorton@nca.nhs.uk (J.G.); david.lewis@nca.nhs.uk (D.L.); rosie.donne@nca.nhs.uk (R.D.); elizabeth.lamerton@nca.nhs.uk (E.L.); rachel.middleton@nca.nhs.uk (R.M.);; 2Faculty of Biology, Medicine, and Health, University of Manchester, Manchester M13 9PL, UK; saira.anwar@student.manchester.ac.uk (S.A.); amnah.ahmed@student.manchester.ac.uk (A.A.)

**Keywords:** COVID-19 vaccination, health inequalities, ethnicity, social deprivation, haemodialysis, peritoneal dialysis, kidney transplant

## Abstract

Background: COVID-19 vaccination has changed the landscape of the COVID-19 pandemic; however, decreased uptake due to vaccine hesitancy has been observed, particularly in patients from minority ethnic backgrounds and socially deprived areas. These patient characteristics are common in patients on Renal Replacement Therapy (RRT), a population at extremely high risk of developing serious illness from COVID-19 and who would thus benefit the most from the vaccination programme. We designed a bespoke COVID-19 vaccination programme for our RRT population with the aim of decreasing health inequalities and increasing vaccination uptake. Methods: Key interventions included addressing vaccine hesitancy by deploying the respective clinical teams as trusted messengers, prompt eligible patient identification and notification, the deployment of resources to optimise vaccine administration in a manner convenient to patients, and the timely collection and analysis of local safety and efficacy data. First, COVID-19 vaccination data in relation to ethnicity and social deprivation in our RRT population, measured by the multiple deprivation index, were analysed and compared to uptake data in the total regional adult clinically extremely vulnerable (CEV) population in Greater Manchester (GM). Univariate logistic regression analysis was used to explore the factors associated with not receiving a vaccine. Results: Out of 1156 RRT patients included in this analysis, 96.7% received the first dose of the vaccination compared to 93% in the cohort of CEV patients in the GM. Age, gender, ethnicity, and a lower index of multiple deprivation were not identified as significant risk factors for poor first dose vaccine uptake in our cohort. Vaccine uptake in Asian and Black RRT patients was 94.9% and 92.3%, respectively, compared to 93% and 76.2% for the same ethnic groups in the reference CEV GM. Vaccine uptake was 96.1% for RRT patients in the lowest quartile of the multiple deprivation index, compared to 90.5% in the GM reference population. Conclusion: Bespoke COVID-19 vaccination programmes based on local clinical teams as trusted messengers can improve negative attitudes towards vaccination and reduce health inequalities.

## 1. Introduction

The UK’s national immunisation programme for COVID-19, launched in December 2020 in a phased manner, was a major public health intervention leading to the reduction in COVID-19-associated morbidity and mortality. The programme was designed to promptly vaccinate high-risk populations based on age, patients with co-morbidities or on immunosuppressive medications associated with increased risk of severe COVID-19, and health care workers (1). The speed of the vaccination rollout was unprecedented and drastically ameliorated the impact of the pandemic on society and the National Health Care System. However, lower vaccination uptake due to a variety of reasons was particularly noted in patients from select ethnic groups and socially deprived areas [1].

Patients on renal replacement therapy (RRT) are at an increased risk of death from COVID-19 infection [2]. Extremely high death rates were observed in those receiving haemodialysis (HD) during the first wave of the pandemic, likely due to the multiple co-morbidities often present in these patients as well as an increased exposure risk due to their inability to shield, as they needed to attend thrice-weekly life-sustaining treatment at hospital facilities [3]. Kidney transplant recipients (KTR) are at extremely high risk from COVID-19 and were particularly affected when shielding precautions were relaxed, exposing them to high community transmission rates [4]. A higher death rate compared to the general population was also observed in patients receiving peritoneal dialysis (PD) treatment [5]. Consequently, RRT patients were characterised as “clinically extremely vulnerable” and prioritised for COVID-19 vaccination.

The population characteristics associated with vaccine hesitancy are common in patients receiving RRT. Social deprivation is associated with the prevalence of chronic kidney disease [6], increased risk of late presentation to renal services [7], and higher incidence of renal replacement therapy [8]. Similarly, patients from South Asian and Black ethnic minority backgrounds are more likely to suffer from conditions leading to chronic kidney disease and, therefore, to start RRT [9]. In addition to the expected vaccine hesitancy due to the above-mentioned factors in the RRT population, patients on HD may find it cumbersome to attend medical facilities outside of their regular HD treatments, which may adversely affect vaccine uptake via shared community pathways that require additional visits to medical facilities.

To this end, we designed a bespoke COVID-19 vaccination rollout programme aimed at increasing total vaccination coverage and reducing the health inequalities in our RRT population.

## 2. Materials and Methods

### 2.1. Setting and Population

Salford Renal Service, part of the Northern Care Alliance Group, provides nephrology outpatients, HD, PD, and KTR care to residents of Salford and the northwestern portion of greater Manchester, a population of ~1.5 m. HD is provided via a large central hub unit (Salford) and spoke units based in stand-alone community sites (Wigan and Oldham) or at hospital site-based units (Salford Royal, Bolton, and Rochdale).

#### 2.1.1. Vaccination Programme Design

Salford hospital was designated as a vaccination hub from the start of the vaccination programme by repurposing the Mayo building, which had previously been used for meetings, administration, and education. Online meetings of the multidisciplinary team (MDT), including renal managers, clinicians, pharmacists, and nurses, were instituted to facilitate rapid vaccination in the eligible patient cohorts and promote vaccination uptake via the ease of availability, education, and reassurance for hesitant populations. The following key areas were identified: identifying eligible patients, addressing vaccine hesitancy using local data and trusted messengers, and allocating resources to optimise vaccine administration in a manner convenient for the patients.

#### 2.1.2. Identification of Eligible Patients

The first task was the identification of patients who were eligible for vaccination due to their belonging to the clinically extremely vulnerable group. The patient identification process included the generation of clinic lists by administration staff, followed by sense–checking by clinicians and nursing teams familiar with the patients and their current and previous treatments. Identified patient groups were added to the national database (NHS High-Risk Digital Database), and letters of eligibility for vaccination were issued to GPs, patients, and local vaccine teams for the primary course and subsequent doses.

### 2.2. Multidisciplinary Clinical Teams as Trusted Messengers across the Renal Replacement Therapy Modalities

In each part of the service, MDT members (doctors, nurses, and administrative staff) who had established rapport with the patients were assigned to discuss the need for and encourage vaccination, with a particular focus on vaccine-hesitant individuals.

#### 2.2.1. Kidney Transplant Recipients (KTR)

All patients had a ‘vaccine conversation’ with a trusted member of the kidney transplant team prior to vaccination addressing any questions and concerns. Bespoke vaccination sessions were offered to all patients at the onsite hospital vaccination hub, recognising the vulnerability of this population both physically and psychologically, many of whom continue to shield themselves from the rest of society.

#### 2.2.2. Haemodialysis (HD)

Pop-up vaccination teams, including HD consultants and renal nurses, were deployed in the HD satellite units, recognising the inconvenience of additional hospital visits for HD patients who attend regular HD treatment three times weekly. All patients had a ‘vaccine conversation’ with the vaccinating team, and patients who declined vaccination were referred to their named consultant for further discussion. Bespoke vaccination sessions were also offered at the onsite hospital vaccination hubs.

#### 2.2.3. Peritoneal Dialysis (PD)

The in-house nursing team contacted each of the PD patients individually by telephone to encourage the initial vaccination and arrange appointments at the onsite hospital vaccination hub, coordinated with scheduled clinic attendance if possible. Any concerns or uncertainty about the vaccine were addressed by the nursing team during community visits or by telephone and by the PD Consultants by telephone or during in-clinic appointments.

### 2.3. Real-Time Collection of Safety and Efficacy Data

Clinicians, medical students, and nursing and administrative staff combined to audit the vaccine administration, collect patient feedback on side effects and tolerability, and monitor the efficacy of the new vaccines.

#### 2.3.1. Safety Data

A questionnaire-based vaccine reactogenicity audit based on the US Food and Drug Administration tool (https://www.fda.gov/media/73679/download, accessed on 15 April 2022) was conducted to assess the safety profile and patient experience of the first cohort of vaccinated HD patients. Briefly, this assessed local and systemic side-effects of the vaccines using a 5-point scale (0—no side effects; 1—mild, does not affect daily activity; 2—moderate, interferes with daily activity; 3—prevents daily activity; 4—requires A&E/hospitalization) (Appendix A). Questionnaires were completed by the patients alone or assisted by medical students or the nursing staff. Local safety data were analysed by the MDT team, and the results were shared with the patients in the form of informal discussions, posters in clinical areas, and social media.

#### 2.3.2. Efficacy Data

Vaccine efficacy was evaluated based on SARS-CoV-2 antibody status, measured during routine blood tests, and COVID-19-related morbidity and mortality extracted from the electronic patient records. Efficacy results were summarised in rapidly developed manuscripts to enhance knowledge and confidence about the vaccination process based on local experience.

### 2.4. Bespoke Vaccine Administration and Support with Transportation

Each team organised initiatives to optimise patient convenience in the timing and location of vaccinations and address concerns related to the potential exposure to the virus in vaccination venues. Vaccination booking for RRT patients was managed by the multidisciplinary team. Transport with taxis was provided to patients who were unable to commute or who had fears of using public transport due to infection risk. Pop-up vaccination teams were deployed to the HD units to offer vaccinations at the time of dialysis.

#### 2.4.1. Resources

All registered staff (medical, nursing, and pharmacy) involved in the vaccination process completed national e-Learning training specific to each COVID-19 vaccine available, in addition to the general immunisation and vaccination training provided by the Trust. Anaphylaxis training and basic life support were revisited, and injection techniques revalidated. Wider staff engagement with the vaccine was delivered through online meetings with a particular focus on the vaccine-related questions raised by patients. Dedicated administrative staff organised the vaccination appointments and database of vaccine uptake and hesitancy. Renal staff supported bespoke sessions over the weekends. MDT teams were involved in live data analysis throughout the vaccination programme.

No vaccination-specific national or private funds were used to support the programme. The additional activity required for the vaccination programme was provided by the existing multidisciplinary team members as overtime work or through the temporary redeployment, and the resulting pressure within the service was covered by overtime from other members of the team. Staff went above and beyond to support the programme. Departmental transport funds were used to cover the cost of hospital travel.

#### 2.4.2. Patient Involvement

Local and regional patient representatives were involved in the vaccination workstream at regular online meetings, provided feedback, and shared emerging evidence using social media platforms and patient to patient discussions. Ethnic minority patient groups were specifically targeted. Presentations were organised for the Greater Manchester Kidney Information Network (https://kinet.site/gmkin/, accessed on 15 April 2022) and the African–Caribbean community initiative for kidney patients WSH BME Network (https://www.wshbmenetwork.org.uk/, accessed on 15 April 2022).

### 2.5. Data Collection, Analysis and Ethical Approval

Demographic and clinical data were extracted from the electronic patient records and tolerability data from the audit questionnaires. Vaccination data were collected until the end of December 2021. Social deprivation data were calculated using the postal code-derived 2019 Index of Multiple Deprivation (IMD) income domain data in deciles from the Ministry of Housing, Communities, and Local Government website [10]. Aggregate vaccination uptake data for the first dose, in relation to ethnicity and social deprivation from the reference Greater Manchester adult population of clinically extremely vulnerable patients, were extracted with permission from the Greater Manchester Tableau on 3 March 2022 (Data Sources Vaccinations Feed (from Arden and Gem CSU), Population Data: Master Patient Index, accessed online from https://www.gmtableau.nhs.uk/#/site/GMHSCPPublic/explore, accessed on 15 April 2022).

A flowchart of patients included in the study is illustrated in Figure 1.

#### 2.5.1. Statistical Analyses

In the descriptive part of the analysis, categorical variables were expressed as numbers (percentage), and continuous variables were expressed as median (interquartile range). Univariate logistic regression analysis was used to explore the factors associated with not receiving a vaccine. All analyses were carried out by SPSS software (version 24), registered to the University of Manchester.

#### 2.5.2. Ethical Approval

This study was registered with the Northern Care Alliance Research and Innovation department (Ref. No.: S21HIP08 and S21HIP51). Due to this being an observational study with complete anonymisation of patient identification details, there was no indication to consent for each individual patient in this study.

## 3. Results

### 3.1. MDT Mobilisation, Safety and Efficacy Data Analysis, and Sharing

MDT engagement and additional resource allocation were successful, and the vaccination workstream was embedded in the core activity across the service. Analysis of the reactogenicity questionnaire audit of the first 144 patients showed a similar side-effect profile for these vaccines in the dialysis population to the published trial data (Appendix A). The results were summarised in posters designed for each clinical area (poster example presented in Figure 2) to engage with patients’ concerns and offer ‘real–world’ local data that would be reassuring. Continuous variables are expressed as median (IQR), and categorical variables are expressed as numbers (percentage).

Local data regarding COVID-19 morbidity and mortality and vaccine efficacy were analysed by the respective MDT teams for each RRT cohort; results were shared with patients, the Trust, and the national vaccination team and were summarised in published manuscripts [4,11,12,13]. In total, 18 doctors, 1 pharmacist, 9 nurses, and 1 member of the administrative staff co-authored publications supported by the wider MDT team.

### 3.2. COVID-19 Vaccine Uptake

From a total of 1212 patients receiving renal replacement therapy, 1156 patients with available postal code, ethnicity, and vaccination data were included in this analysis. The median age of our cohort was 58 years, with a predominance of males (62.4%) and Caucasians (78.2%). A total of 68% had a history of hypertension, 27.6% were diabetic, and 25% had a previous history of cardiovascular disease. A major proportion of our cohort was in the lowest quintile (most deprived) of the index of multiple deprivation (44.8%) (Table 1). Vaccine uptake, based on ethnic background and the multiple deprivation index in RRT patients and CEV patients in Greater Manchester, is presented in Table 2. Of the total RRT cohort, 96.7% received the first dose of the vaccination, compared to 93% in the cohort of adult clinically extremely vulnerable patients in the Greater Manchester cohort. Higher percentages of first dose vaccine uptake were observed in our RRT population in Asian (94.9%) and Black (92.3%) subjects compared to the reference clinically extremely vulnerable population (93% and 76.2%, respectively). Vaccine uptake across all ethnic groups in RRT patients and clinically extremely vulnerable patients in Greater Manchester is presented in Table 2. A similar observation was observed across the quintiles of the index of multiple deprivation, with 96.1% of RRT patients in the lowest quartile of the multiple deprivation index having received the first dose of the vaccine compared to 90.5% in the Greater Manchester reference population.

In the univariate binary logistic regression analysis, age, sex, ethnicity, and the index of multiple deprivation were not identified as significant risk factors for the non-uptake of the first dose vaccine (Table 3).

## 4. Discussion

Our results show that a bespoke, adequately resourced vaccination strategy using local evidence and clinical teams as trusted messengers led to increased overall COVID-19 vaccine uptake and reduced ethnic and social deprivation inequalities in our RRT patients compared to the reference clinically extremely vulnerable adult population of Greater Manchester.

Vaccine hesitancy has been more prevalent in minority ethnic groups and patients from socially deprived backgrounds, despite increased COVID-19-related morbidity and mortality. [14]. It has been speculated that, although low health literacy may play a role, hesitancy is mainly driven by mistrust in these patient groups towards drug industry research, healthcare systems, and government policies due to historical structural inequalities [14]. The infamous US Tuskegee syphilis study in 1993, which deprived Black participants of antibiotic treatment when it became available to inspect the natural history of untreated diseases [15], is exemplary in the history of unethical medical research that continued into the 1990s in some academic institutions [15]. The rapid development of COVID-19 vaccines with the underrepresentation of ethnic minority groups in the vaccine clinical trials [16], combined with the intensification of structural health inequalities during the pandemic that amplified mistrust in public health policies, were factors that contributed to increased hesitancy to novel COVID-19 vaccines.

Our results show that the local trust relationship between the clinical team and RRT patients can address vaccine hesitancy to a substantial proportion, in line with previous research demonstrating that RRT patients from ethnic minorities who exhibit mistrust toward the health care system as a whole can establish relationships of trust with selected elements of the health care system with whom they have direct and frequent contact [17]. Our results are also in line with a recently published study showing that local autonomy for targeted initiatives can improve outcomes in minority groups [18].

The investment of resources for local data collection, analysis, and early dissemination was particularly important in consolidating the relationship of trust between the local clinical team and the patients. When the MDT team shared the results showing reduced vaccine efficacy following the first dose in KTR patients [4] with their patients, at odds with the general perception of reassuring vaccine efficacy in immunosuppressed patients at that point in time [19], the initiative to analyse and share the data was welcomed by the patients, confirming that local knowledge, openness, and transparency are essential to building and maintaining the relationship of trust with patients.

The successful COVID-19 vaccination programme can inform initiatives addressing inequalities related to patients’ hesitancy for other important vaccination programmes (flu and hepatitis B) as well as in developing focused programmes in other areas of health care inequalities that are particularly prominent in this cohort of patients. Securing additional resources to enable the delivery of bespoke local programmes is of paramount importance and should be considered carefully in future funding plans for renal services.

The following limitations of our study should be considered: We did not have access to vaccination uptake RRT data from other centres with different vaccination strategies, and it is unknown if these patients may have exhibited different hesitancy behaviour compared to the wider population of clinically extremely vulnerable patients. However, a preprint version of a study in 4697 dialysis patients from the US with 23.4% Black participants, representative of the US dialysis population, showed only a 76% COVID-19 vaccine uptake by September 2021 for at least one dose of a COVID-19 vaccine [20]. Finally, our access to the Greater Manchester clinically extremely vulnerable adult population was limited to aggregate data, and we could not perform statistical comparisons with our patients.

## 5. Conclusions

Bespoke COVID-19 vaccination programmes based on MDT targeted initiatives and driven by local evidence can improve negative attitudes toward vaccination. Adequatelyresourced, bespoke programmes using the clinical team as trusted messengers to address hesitancy-related health inequalities in RRT patients should be prospectively investigated.

## Figures and Tables

**Figure 1 vaccines-10-00939-f001:**
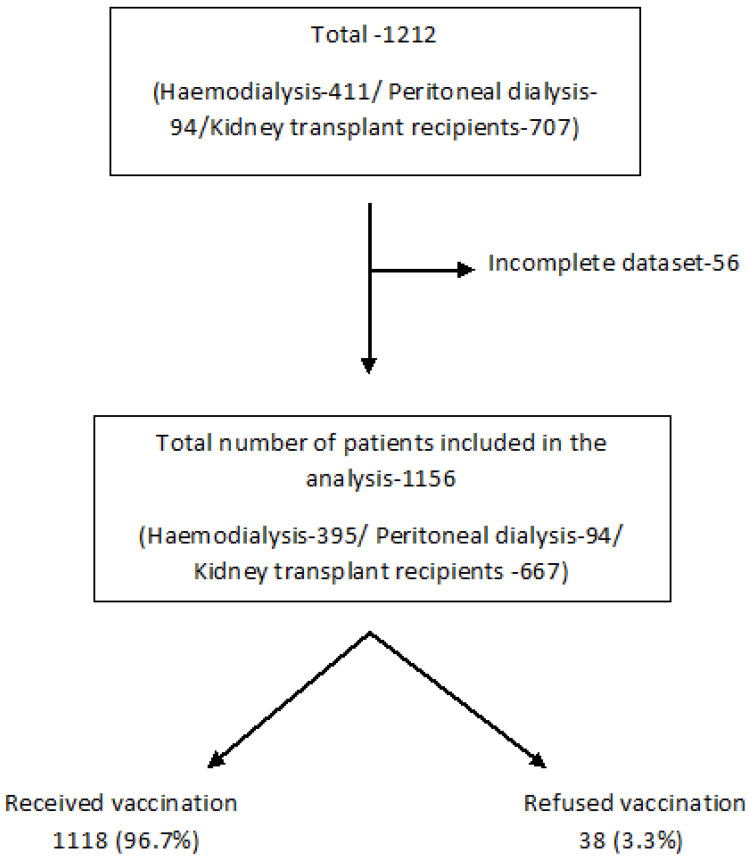
A flowchart of patients.

**Figure 2 vaccines-10-00939-f002:**
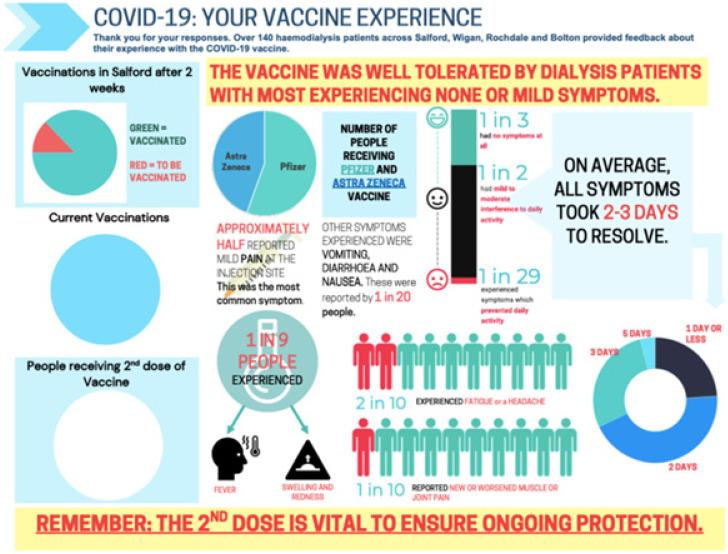
Poster example summarising the results of the local safety audit of 144 patients after receiving the first dose of COVID-19 vaccination.

**Table 1 vaccines-10-00939-t001:** Distribution of characteristics of the sample population.

Variables	Total 1156	Haemodialysis 395	Peritoneal Dialysis 94	Kidney Transplant 667
Age	58 (47–68)	62 (50–73)	64 (48–72)	56 (46–64)
Gender (male)	721 (62.4)	252 (63.8)	58 (61.7)	411 (61.6)
Ethnic domain White Asian Black Other	904 (78.2) 198 (17.1) 39 (3.4) 15 (1.3)	276 (69.9) 92 (23.3) 21 (5.3) 6 (1.5)	79 (84) 13 (13.8) 1 (1.1) 1 (1.1)	549 (82.3) 93 (13.9) 17 (2.5) 8 (1.2)
Diabetes mellitus	319 (27.6)	195 (49.4)	32 (34)	92 (13.8)
Hypertension	787 (68.1)	285 (72.2)	79 (84)	423 (63.4)
Cardiovascular disease	287 (24.8)	114 (28.9)	37 (39.4)	136 (24.6)
Index of Multiple Deprivation (Quintiles) Most deprived Q1 Q2 Q3 Q4 Least deprived Q5	518 (44.8) 219 (18.9) 140 (12.1) 160 (13.8) 119 (10.3)	231 (58.5) 70 (70.7) 38 (9.6) 33 (8.4) 23 (5.8)	44 (46.8) 14 (14.9) 14 (14.9) 13 (13.8) 9 (9.6)	243 (36.4) 135 (20.2) 88 (13.2) 114 (17.1) 87 (13)
Received first dose vaccination	1118 (96.7)	383 (97)	92 (97.9)	643 (96.4)

**Table 2 vaccines-10-00939-t002:** Comparison of first dose vaccination uptake based on ethnicity and index of multiple deprivation between Salford cohort and Greater Manchester adult cohort.

First Dose Vaccination	Great Manchester Adult CEV Cohort		Salford Cohort		
Ethnicity	Total 75,188 of 80,887 93%	Total 1118 of 1156 96.7%	Haemodialysis 383 of 395 97%	Peritoneal dialysis 92 of 94 97.9%	Kidney Transplant 643 of 667 96.4%
White Mixed Asian or Asian British Black, African, Caribbean, or Black British Other ethnic groups	94% 79.4% 93% 76.2% 85.1%	97.3% 88.9% 94.9% 92.3% 100%	97.8% 85.7% 96.7% 90.5% 100%	98.7% 100% 92.3% 100% 100%	96.9% 100% 93.5% 94.1% 100%
Index of Multiple Deprivation (Quintiles)					
Most deprived Q1 Q2 Q3 Q4 Least deprived Q5	90.5% 93.2% 95.2% 96.2% 97.2%	96.1% 97.3% 97.1% 95.6% 99.2%	95.7% 100% 94.7% 97% 100%	97.7% 100% 92.9% 100% 100%	96.3% 95.6% 97.7% 94.7% 98.9%

CEV: Clinically extremely vulnerable adult population.

**Table 3 vaccines-10-00939-t003:** Univariate logistic regression analysis: odds of not receiving a vaccine.

Total (Data Available)	OR (95%CI)	*p*-Value
Age	0.99 (0.96–1.01)	0.242
Male gender	1.32 (0.66–2.6)	0.435
Caucasian ethnicity	0.52 (0.26–1.04)	0.064
Index of multiple deprivation quintiles	0.88 (0.69–1.21)	0.303

OR–odds ratio, CI–confidence interval.

## Data Availability

The data analysed in the current study are available from the corresponding authors on reasonable request.

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
