# Peer review of "Increasing Uptake of COVID-19 Vaccination and Reducing Health Inequalities in Patients on Renal Replacement Therapy—Experience from a Single Tertiary Centre"

_vaccines, 2022, doi:10.3390/vaccines10060939_

Round 1

Reviewer 1 Report

An excellent, well organized, successful initiative to improve the COVID-19 vaccination of patients on RRT. A great example for clinicians to refer to and utilize in setting up their own programs. 

Author Response

Thank you very much

Reviewer 2 Report

Thank you for the opportunity to review this very interesting paper.

Authors demonstrated that a bespoke adequately resourced vaccination strategy using local evidence and the clinical teams as trusted messengers led to increased overall COVID-19 vaccine uptake and reduced ethnic and social deprivation inequalities

 The pandemic has intensified social inequalities, particularly among high-risk subjects for severe disease.

The promotion of vaccines and the implementation of strategies to increase vaccination coverage are a concrete contribution to the fight against the pandemic.

 The paper is well written and easy to understand. I have minor comments about the paper.

It would be interesting to know more details about the training received by the multidisciplinary team about the COVID vaccines before engage patients   and before having a vaccine conversation with patients.

 Also, It would be interesting  to know how many resources were needed to carry out the bespoke sessions and vaccination service  Have dedicated staff been hired? Were national or private funds used for the bespoke sessions (ie transport service, for the working hours of the staff who did the interviews with the patients)?

Minor

 Line 90, page 2 Please, clarify the sentence  ‘Organization and resource optimization directed at the outpatient and dialysis cohort.’

Line 144, page 4: SARS-CoV2 instead of SARS-COV-2

Author Response

The paper is well written and easy to understand. I have minor comments about the paper.

Q1: It would be interesting to know more details about the training received by the multidisciplinary team about the COVID vaccines before engage patients   and before having a vaccine conversation with patients.

Answer: Thank you. We have now included the points below in the methods section to make this aspect clear.

All registered staff ( medical, nursing and pharmacy) involved in the vaccination process completed national e-Learning training specific to each COVID-19 vaccine available in addition to general immunisation and vaccination training provided by the Trust. Anaphylaxis training and basic life support was revisited, and injection techniques revalidated. Wider staff engagement with the vaccine was delivered through online meetings with particular focus on vaccine related questions raised by patients”.

Q2 Also, It would be interesting  to know how many resources were needed to carry out the bespoke sessions and vaccination service  Have dedicated staff been hired? Were national or private funds used for the bespoke sessions (ie transport service, for the working hours of the staff who did the interviews with the patients)?

Answer: Thank you. We have now included the points below in the methods section to make this point clear.

“There were no vaccination specific national or private funds used to support the programme. The additional activity required for the vaccination programme was provided by the existing multi-disciplinary team members as overtime work or through temporary redeployment and the resulting pressure within the service was covered with overtime by other members of the team. Staff went above and beyond to support the programme. The departmental transport funds were used to cover the cost of hospital travel”.

Q3. Line 90, page 2 Please, clarify the sentence ‘Organization and resource optimization directed at the outpatient and dialysis cohort.’

Answer: We have now removed this sentence to improve clarity.

Q4 Line 144, page 4: SARS-CoV2 instead of SARS-COV-2

Answer: Thank you. We have corrected this typo error.

Reviewer 3 Report

I considered the manuscript entitled “Increasing uptake of COVID-19 vaccination and reducing health inequalities in patients on renal replacement therapy- Experience from a single tertiary centre” Dimitrios Poulikakos, et al that is intended to be published in Vaccines journal.

The manuscript deals with a commendable social and health initiative which tried to improve the chance for the vaccination in a special group of patients. The manuscript and the study are well planned, developed and presented.

One major strength is that authors designed a bespoke COVID-19 vaccination program for a RRT population with the aim to decrease health inequalities and increase vaccination uptake. Individual or collective initiatives have an impact on the collective good. On the other hand, a weakness of the study is that given the speed of the topics in Covid, at this time these results are a little out of date. It has been shown that these patients need three and four vaccine shots. And also, at this time, in some countries Evusheld is being introduced given the low immunization response in this community.

No concern with the study and the manuscript. However, results are somewhat modest compared with the general population described: Out of 1156 RRT patients included in this analysis 96.7% received the first dose vaccination compared to 93% in the cohort of CEV patients in the GM. I am not sure the effort performed by the team, greatly increased the rate of vaccination.

For me the study is merely descriptive and there is no comparison with any other group of patients out of the bespoke COVID-19 vaccination program for a RRT, which could suggest the benefits thanks to the effort of the group

Author Response

[Vaccines] Manuscript ID: vaccines-1734515

Point by Pont response to reviewers’ comments

Reviewer 3

Comment: On the other hand, a weakness of the study is that given the speed of the topics in Covid, at this time these results are a little out of date. It has been shown that these patients need three and four vaccine shots. And also, at this time, in some countries Evusheld is being introduced given the low immunization response in this community.

Response: We agree with the reviewer that these patients need focused attention and different vaccination routines. We have recognised and highlighted this need in the articles referenced in the submitted manuscript (12,13,14) and we shared this information with our patients to promote uptake for subsequent vaccinations and advise caution to immunosuppressed patients regarding the relaxation of general IPC measures after the first vaccination dose in England (14).  However, this manuscript aims at exploring uptake in relation to vaccine hesitancy and we used acceptance of the first dose of vaccination as surrogate of overcoming hesitancy to the vaccination programme.

Comment:  However, results are somewhat modest compared with the general population described: Out of 1156 RRT patients included in this analysis 96.7% received the first dose vaccination compared to 93% in the cohort of CEV patients in the GM. I am not sure the effort performed by the team, greatly increased the rate of vaccination. For me the study is merely descriptive and there is no comparison with any other group of patients out of the bespoke COVID-19 vaccination program for a RRT, which could suggest the benefits thanks to the effort of the group.

Response: We have highlighted the limitation of lack of statistical comparison with other bespoke vaccination programmes in RRT populations in the discussion (lines 294-302). We did not have available data to perform such comparison. Our analysis included only internal statistical calculations that did not show any significant difference for the selected variables (age, ethnicity, social deprivation) and this indicates a successful vaccination effort. We also presented aggregate data from the local CEV population and data from a study in the US dialysis population (in pre-print) showing different uptakes based on ethnicity/deprivation as reference data but we could not perform statistical comparisons with our population.